# Association between SNPs in microRNAs and microRNAs-Machinery Genes with Susceptibility of Leprosy in the Amazon Population

**DOI:** 10.3390/ijms231810628

**Published:** 2022-09-13

**Authors:** Mayara Natália Santana da Silva, Diana Feio da Veiga Borges Leal, Camille Sena, Pablo Pinto, Angélica Rita Gobbo, Moises Batista da Silva, Claudio Guedes Salgado, Ney Pereira Carneiro dos Santos, Sidney Emanuel Batista dos Santos

**Affiliations:** 1Laboratório de Genética Humana e Médica, Instituto de Ciências Biológicas, Universidade Federal do Pará, Belém 66075-110, PA, Brazil; 2Laboratório de Biologia e Eletrofisiologia Celular, Seção de Parasitologia, Instituto Evandro Chagas, Ananindeua 67030-000, PA, Brazil; 3Núcleo de Pesquisas em Oncologia, Instituto de Ciências Biológicas, Universidade Federal do Pará, Belém 66073-000, PA, Brazil; 4Laboratório de Dermato-Imunologia, Instituto de Ciências Biológicas, Universidade Federal do Pará, Belém 66075-110, PA, Brazil

**Keywords:** leprosy, SNPs, biomarkers, mycobacteria, host-pathogen interaction, Amazon, genetics population

## Abstract

Leprosy is a chronic neurodermatological disease caused by the bacillus *Mycobacterium leprae*. Recent studies show that SNPs in genes related to miRNAs have been associated with several diseases in different populations. This study aimed to evaluate the association of twenty-five SNPs in genes encoding miRNAs related to biological processes and immune response with susceptibility to leprosy and its polar forms paucibacillary and multibacillary in the Brazilian Amazon. A total of 114 leprosy patients and 71 household contacts were included in this study. Genotyping was performed using TaqMan Open Array Genotyping. Ancestry-informative markers were used to estimate individual proportions of case and control groups. The SNP rs2505901 (*pre-miR938*) was associated with protection against the development of paucibacillary leprosy, while the SNPs rs639174 (*DROSHA*), rs636832 (*AGO1*), and rs4143815 (*miR570*) were associated with protection against the development of multibacillary leprosy. In contrast, the SNPs rs10739971 (*pri-let-7a1*), rs12904 (*miR200C*), and rs2168518 (*miR4513*) are associated with the development of the paucibacillary leprosy. The rs10739971 (*pri-let-7a1*) polymorphism was associated with the development of leprosy, while rs2910164 (*miR146A*) and rs10035440 (*DROSHA*) was significantly associated with an increased risk of developing multibacillary leprosy.

## 1. Introduction

Leprosy is a chronic infectious and contagious dermatoneurological disease caused by *Mycobacterium leprae* and *Mycobacterium lepromatosis*. The bacillus *M. leprae*, the most common causative agent of leprosy, is an intracellular pathogen that has a tropism for macrophages and Schwann cells [1,2]. The skin and peripheral nerves are the main areas affected by the disease, which can lead to sensory loss and muscle atrophy in face, hands and feet, with consequent disability if left untreated.

Despite being considered eradicated in many countries, leprosy remains a major public health problem in countries such as India, Brazil, and Indonesia. In 2020, the World Health Organization (WHO) recorded 202,162 new cases of leprosy in the world, and of the 29,936 cases reported in the Americas region, the equivalent of 93% (27,864) occurred in Brazil [3]. The Pará state, located in the Brazilian Amazon region, is classified as hyperendemic for leprosy according to the National Notifiable Diseases Information System (SINAN) in 2019 [4].

The variability of clinical manifestations of leprosy is intrinsically linked to the host’s immune response against *M. leprae.* depending on the action of the cell-mediated immune response or humoral immune response [5,6,7]. One of the main leprosy classification systems, the Ridley–Jopling classification, is based on immunological, pathological, and clinical criteria categorized into five groups: tuberculoid (TT), borderline tuberculoid (BT), borderline (BB), borderline lepromatous (BL), and lepromatous (LL) [6]. The WHO has an operational classification that uses the number of skin and nerve lesions: paucibacillary (PB) leprosy for patients with lesions equal to or below five, and multibacillary leprosy (MB) for patients with lesions above five [8].

The development of leprosy is associated with several factors such as socioeconomic status, time of exposure to the bacillus, and genetic characteristics related to the host [9,10,11,12]. Recently, the study of miRNAs has advanced to several infectious diseases, including leprosy [13,14,15]. MiRNAs are small non-coding RNA molecules capable of regulating the gene expression of mRNA target molecules, having an important role in the regulation of the innate and adaptive immune response [16,17]. Single-nucleotide polymorphisms (SNPs) in genes related to miRNAs play a significant role in the generation, processing, and function of these molecules, and contribute to multiple phenotypes and diseases [18,19,20,21].

In recent years, genetic variants in miRNA genes and machinery related to miRNA processing have been associated with cancer [18,22,23,24] and infectious diseases [13,25,26,27] including mycobacterial diseases such as tuberculosis and leprosy in different populations. MiRNAs are involved in several important biological processes, such as modulation of the immune and adaptive response against pathogens [28,29]. They can target hundreds of mRNAs and regulate them, which implicates them in cell proliferation, differentiation, apoptosis, and pathways of the immune response [25,30,31]. SNPs in miRNA genes can alter the transcription of the primary miRNA transcript and the interaction of miRNA with mRNA, affecting the expression of certain genes [32].

This study investigated the association of twenty-five genetic variants in miRNAs and miRNA machinery-related genes (*DROSHA* and *AGO1*) with leprosy susceptibility in a population from the Amazon region, northern Brazil.

## 2. Results

### 2.1. Clinical and Demographic Characteristics of the Groups Studied

Clinical and demographic data such as age, sex, and genetic ancestry of the case and control groups were compared to verify possible confounding factors (Table 1). We did not observe statistically significant differences in the variables age and sex, although our data demonstrate that the mean age of patients was higher in the case group, and men were more frequent among the patient group, while women were more frequent in the control group. The population structure showed that the mean frequency of European ancestry was higher in the group of patients (*p* = 0.007), while the mean frequency of African ancestry was higher in the group of healthy individuals (*p* = 0.012) (Table 1).

Table 1 also summarizes the clinical and demographic characteristics of leprosy patients grouped according to paucibacillary (PB) and multibacillary (MB) clinical forms. We observed a significant difference between age, with a higher mean among MB patients, and gender, with a greater number of male patients in the MB group and a greater number of female patients in PB. This result corroborates with several other studies on different populations and WHO data [3,4,7,13]. Concerning the age between patients from different leprosy groups, the difference is expected due to the longer time to develop MB leprosy. When comparing the clinical forms of leprosy, we did not observe any statistical differences regarding the proportions of genetic ancestry (Table 1).

### 2.2. Analysis of Association of miRNA and DROSHA Precursor Genes with Leprosy Susceptibility

Twenty-five SNPs were analyzed, and twenty-one were in Hardy–Weinberg Equilibrium in the case and control groups (*p*-value > 0.05); therefore, they were included in the genotypic association analyses, while the markers that presented HWE deviation in one of the studied groups were excluded from further analyses.

Table 2 summarizes the significant results found in this study according to the genotypic associations between the case group of leprosy patients versus healthy control group, paucibacillary leprosy patients versus control, multibacillary leprosy patients versus control, and finally, patients grouped according to clinical form PB versus MB.

In the association analysis between leprosy patients and healthy individuals, we found six statistically significant markers for the risk of developing leprosy in the population studied: rs2505901 (*pre-mir938*), rs639174 (*DROSHA*), rs636832 (*AGO1*), rs10739971 (*pri-let -7a1*), rs12904 (*miR200C*) and rs10035440 (*DROSHA*) (Table 2). The complete results can be viewed in Appendix A.

According to the results, the SNP rs2505901 (*pre-mir938*) was associated with a decreased risk of leprosy in both the dominant (*p* ≤ 0.01; OR = 0.40; 95%CI = 0.20–0.81) and recessive model (*p* = 0.03 OR = 0.41; 95%CI = 0.18–0.95), showing that the CC and CT genotype can protect against the disease. SNPs rs639174 (*DROSHA*) and rs636832 (*AGO1*) were also associated with decreased risk of leprosy in the dominant model (*p* = 0.02; OR = 0.45; 95%CI = 0.23–0.89 and *p* = 0.01; OR = 0.45; 95%CI = 0.23–0.89, respectively).

The marker rs10739971 (*pri-let-7a1*) was associated with increased risk of leprosy in analyzes using the dominant (*p* = 0.02; OR = 4.66; 95%CI = 1.17–18.6) and recessive model (*p* = 0.04; OR = 5.09;95%CI = 0.95–7.45). Likewise, SNPs rs12904 (*miR200C*) and rs10035440 (*DROSHA*) showed a positive association with the risk of disease development in the dominant model (*p* = 0.01; OR = 2.77; 95%CI = 1.25–6.11 and *p* = 0.03; OR = 2.04; 95%CI = 1.03–4.02, respectively).

Comparing leprosy patients grouped according to the paucibacillary clinical form (PB) with the control group, we found a significant association in SNPs rs2505901 (*pre-miR938*), rs10739971 (*pri-let-7a1*), rs12904 (*miR200C*) and rs2168518 (*miR4513*) (Table 2). The marker rs2505901 (*pre-miR938*) was associated with a decreased risk of PB leprosy in recessive (*p* ≤ 0.01; OR = 0.31; 95%CI = 0.12–0.75) and dominant models (*p* = 0.01; OR = 0.31; 95%CI = 0.12–0.84). On the other hand, SNPs rs10739971 (*pri-let-7a1*) and rs12904 (*miR200C*) were associated with increased risk of paucibacillary leprosy in a dominant model (*p* ≤ 0.01; OR = 13.3; 95%CI = 1.82–90.0 and *p* ≤ 0.01; OR = 3.24; 95%CI = 1.36–8.57) and rs2168518 (*miR4513*) in recessive model (*p* = 0.03; OR = 7.68; 95%CI = 0.83–70.5) (Appendix A).

Comparing genotypes of patients grouped according to multibacillary (MB) clinical form with the control group, SNPs rs639174 (*DROSHA*), rs636832 (*AGO1*) and rs4143815 (*miR570*) were associated with a reduced risk of MB leprosy using a dominant model (*p* = 0.03; OR = 0.43; 95%CI = 0.19–0.98, *p* = 0.04; OR = 0.45; 95%CI = 0.21–0.99 and *p* = 0.03; OR = 0.45; 95%CI = 0.21–0.96, respectively) (Table 2). On the other hand, SNP rs10035440 (*DROSHA*) was associated with increased disease risk in a dominant model (*p* = 0.01; OR = 2.88; 95%CI = 1.22–6.79). The complete data can be viewed in Appendix A.

In a genotype comparison analysis between PB and MB patients, only SNP rs10739971 (*pri-let-7a1*) showed an association with increased risk for leprosy in the PB form in a dominant model (*p* = 0.01; OR = 5.21; 95%CI = 1.59–7.86) (Table 2). While the rs2910164 SNP (*miR146A*) was associated with a decreased risk of leprosy in the PB form in a recessive model (*p* = 0.04; OR = 0.14; 95%CI = 0.01–1.36) (Appendix A).

## 3. Discussion

It is known that the exact mode of leprosy transmission is still not well understood; however, continuous exposure may result in an increased risk of infection, and genetic factors play important roles in the host immune response against *Mycobacterium leprae* [7,9,10]. Thus, using contacts in genetic association studies in leprosy will tell us more adequately whether the polymorphisms studied may be an important factor in disease susceptibility [33]. In this study, we chose to use only household contacts with negative anti-PGL-1 and PCR as the healthy control group. In Pará State, the location in the Brazilian Amazon where the study was conducted, there is no adequate data on hot spots, and the technical data from the Brazilian Ministry of Health recognizes the entire geographical extent as a region of leprosy hyperendemicity, with an annual prevalence rate of ≥ 20 cases per 10,000 inhabitants [4]. Therefore, the population groups included in this study live in a region of high endemicity.

The Brazilian population is highly mixed in terms of the genetic contribution of different continental groups, being composed mainly of Europeans, Amerindians, and Africans. In case–control association studies, genetic ancestry has high relevance in mixed populations, since it can influence the genotypic distribution, due to population stratification. Our data show that the contribution of different ethnic groups in the genetic composition of the Amazonian population can influence the risk of developing leprosy, with the European ethnic contribution being greater in the group of patients and the African ethnic contribution, in the opposite way, having a higher frequency in the control group. This result corroborates with a study carried out by Pinto et al., (2015), it was found that the increase in the European interethnic contribution increases the risk of developing leprosy, while the increase in the African contribution decreases the risk of developing the disease in the Amazonian population [34]. Another study, conducted with patients infected with *M. tuberculosis*, also demonstrated the great importance of genetic ancestry in the development of mycobacterial diseases among Amazonian populations [35].

Currently, there are few studies on the association of polymorphisms in miRNA genes and factors that act via miRNA machinery in leprosy and infectious diseases in general in the Amazon region. In our work, the TT/CT genotypes of the SNP rs2505901 (*pre-miR938*) were associated with protection against the development of leprosy per se and in the PB form. SNP-like variants of the *pre-miR938* gene have been associated with changes in miR938 biogenesis and stability [36]. According to genotype expression data from the GTEx portal, rs2505901 TT/CT (*pre-miR938*) has decreased expression about the CC genotype, showing that the TT/CT genotypes downregulate miR938 expression [37]. MiR938 is associated with regulatory pathways related to cell survival and apoptosis [36,38].

Furthermore, the inflammatory cytokines IL-6 and IL-17A are potential targets of miR938. IL-6 is a cytokine with a pleiotropic activity that acts in the acute inflammatory response and activation of Th17 lymphocytes, also inhibiting pro-inflammatory and immunosuppressive T cells [39]. While IL-17A is produced by CD4+ Th17 cells, it is involved in neutrophilia, inflammation, tissue destruction, and repair through the control of regulatory molecules (programmed death-1/programmed death ligand-1) and is also related to reverse reaction (RR) episodes in leprosy patients [40,41,42]. In a study carried out by Sadhu et al., the frequency of Th17 cells (CD4, CD45RO, IL-17) was significantly higher in BT/TT patients [43]. Santos et al., found higher concentrations of IL-17A in lesions from TT patients and serum from PB patients when compared to LL and MB patients, respectively, associating Th17 cells with the inflammatory response in PB patients [44].

The rs639174 variant (*DROSHA*) is an intronic SNP with a recognized role in transcriptional regulation, and in our study, the CC genotype of this polymorphism, in a dominant model, was associated with protection against leprosy per se and MB. Interestingly, the TT genotype of the SNP rs10035440 (*DROSHA*), which also plays an important role in the splicing and transcriptional regulation of the *DROSHA* gene [45], was associated with the risk of developing leprosy per se and MB in the dominant model. Investigating the effect of these polymorphisms on the GTEx portal, the CC genotype of rs639174 (*DROSHA*) can decrease gene expression compared to other genotypes, while the TT genotype of rs10035440 (*DROSHA*), conversely, can increase gene expression [37].

The *DROSHA* gene encodes a type III RNase and a subunit of the eponymous microprocessor complex, which catalyzes the initial processing step of pri-miRNAs, producing pre-miRNAs [46]. Previous studies have associated rs10035440 (*DROSHA*) with the high risk of developing tuberculosis in the Amazonian population [47].

The genetic variant rs636832 (*AGO1*) was associated with protection against the development of MB leprosy. This SNP is located in the intron region of the gene, so the proposed mechanism of effect for this variant is its potential influence on mRNA splicing or via the activities of intronic regulatory elements [48,49]. The *AGO1* gene encodes a member of the Argonaute family of proteins, which can associate with small RNAs and play an important role in RNA interference and silencing and transcriptional regulation of target genes. AGO1 inhibits cell proliferation by inducing apoptosis, in addition to regulating genes that influence cell cycle growth, survival, and progression [22,50,51].

Our results showed that the SNP rs10739971 (*pri-let-7a1*), in a dominant model, was shown to be associated with the development of leprosy per se and the PB form, with an association between groups of patients (PB versus MB), with increased risk to the PB form. The miRNAs let-7 family plays important roles in several biological processes, including inflammation, immunity, cell proliferation, and differentiation [52,53]. In a miRNoma expression analysis, hsa-let-7f-5p was downregulated in LP in lesions and blood of leprosy patients [54]. In another study performed by Kumar et al., the miRNA profile of macrophages infected with Mycobacterium tuberculosis showed downregulation of miR-let-7f [55]. This microRNA targets the A20 protein, an inhibitor of the NF- kB pathway, so let-7f expression decreases and A20 increases with the progression of *M. tuberculosis* infection in mice [56]. Wambier et al., report that PB leprosy patients, who present greater immune reactivity against *M. leprae*, exhibited less NF- kB activation when compared to MB patients [57].

Our data demonstrate that the rs2910164 (*miR146A*) polymorphism was associated with decreased risk of PB leprosy and consequent susceptibility to risk of MB leprosy. This genetic variant is located in the middle of a stem hairpin, suggesting that these SNPs in pre -miRNAs can alter the secondary structure conformation, and consequently alter the expression of mature miRNA [58,59]. According to a study by Shen et al., this variation from G to C in miR146A resulted in an elevated expression of the mature miRNA when compared to the common allele [59]. The SNP rs2910164 (*miR146A*) has also been associated with leprosy in a population in southeastern Brazil [60]. Regarding mycobacterial diseases, Li et al., reported that the rs2910164 variant (*miR146A*) plays different roles in two distinct ethnic populations, with the G allele increasing the risk of pulmonary tuberculosis in the Tibetan population, while the C allele increases the risk of the disease. in a Han population [61]. However, in a study carried out with the Iranian population, there was no significant association between rs2910164 (*miR146A*) and the risk of tuberculosis [62].

In our study, we suggest the rs4143815 variant (*miR570*) is associated with protection against MB leprosy and, according to the expression database (GTEx), decreases miR570 expression [36]. MiR570 was initially identified in airway epithelial cells, involved in the regulation of the inflammatory response [63]. In a study carried out by Roff et al., it was observed that miR570 can increase the expression of CCL4 and CCL5 and, at the same time, inhibit the expression of CCL2 after a strong inflammatory stimulus, indicating a complex system of direct and indirect regulation [64]. Chemokines play an important role in granuloma formation in diseases caused by mycobacteria. CCL2 is a chemokine capable of recruiting monocytes, memory T cells, and dendritic cells to sites of tissue injury and infection, suggesting maintenance of granuloma integrity in asymptomatic patients. Several studies report that the varying levels of chemokines such as CCL2 can influence the predisposition and severity of leprosy [33].

The rs12904 variant (*miR200C*), in a dominant model, was associated with leprosy per se and PB form. The miR200c gene encodes a member of the miR200 family, which is known to play an important role in epithelial–mesenchymal transition (EMT) [65,66,67]. EMT is a biological process responsible for causing polarized epithelial cells that interact with the basement membrane to lose their intracellular adhesion and acquire a mesenchymal cell phenotype, increasing the capacity for migration, invasion, resistance to apoptosis, and inducing fibrosis [68,69,70]. According to a study by Salgado et al., miRNAs ZEB1/2, which are transcriptional repressors of the miR200 family, were downregulated in LL patients. SOX miRNAs were upregulated in LP, indicating that ZEB1/2 may be a regulator of SOX2 expression [54].

The rs2168518 polymorphism (*miR4513*), in a recessive model, is associated with the development of the PB form of leprosy. miR-4513 is related to cell proliferation, invasion, and EMT and was recently reported to be overexpressed in cancer cell lines [71,72,73,74]. According to work carried out by Xu et al., downregulation of miR-4513 inhibited cell proliferation, migration, and invasion, and at the same time promoted apoptosis [75]. Some polymorphisms in the miR-4513 seed sequences are critical regions for target binding specificity of miRNAs [76]. The rs2168518 variant (*miR4513*) is an example of a polymorphism that is inserted in the seed region of hsa-mir-4513, affecting the regulation of this miRNA. Previous studies have reported the association of this variant with fasting glucose, lipid traits [73], disease risk hitchhiker [75,77], lung adenocarcinoma [78] and age-related macular degeneration [79].

There are very few works in the literature that address the role of miRNAs in relation to leprosy. This is the first study to report how SNPs in miRNA genes and miRNA processing-related machinery may be associated with a predisposition to the development of leprosy and its different clinical forms, PB and MB, especially in an understudied population, such as admixed Brazilian Amazonian population. In the present study, we demonstrated a strong association with the risk of developing leprosy and its different forms in genetic variants in the miRNA genes *pre-miR938, miR570, pri-let-7a1, miR200C, miR4513, miR146A* and variants in machinery-related genes of *DROSHA* miRNAs and *AGO1* in the Amazonian population. The findings may provide valuable information for a better understanding of how genetic factors influence the pathophysiology of the disease, with the potential to find predictive markers of the development of leprosy in this population.

## 4. Materials and Methods

### 4.1. Sampling

The case group consisted of 114 individuals diagnosed with leprosy in the Dr. Marcello Cândia Reference Unit in Sanitary Dermatology of the State of Pará (URE). The populations attended in this reference medical clinical are in a condition of socioeconomic vulnerability. Patients were grouped according to clinical form, totaling 56 patients with paucibacillary leprosy (BT and TT: PB) and 58 with multibacillary form (BL and LL: MB). For the control group, 71 samples were included from individuals who lived or had close contact with leprosy patients, had no clinical signs of leprosy after being examined by experienced leprologists, and were negative for qPCR and anti-PGL-I IgM serology. All the individuals included in the study were residents of Pará, a state located in the Amazon region of Brazil and a hyperendemic leprosy area [4]. Thus, the individuals in the control group are exposed to the same environmental conditions and bacterial load, are intra- or peridomestic contacts of leprosy patients and are not related to the case group of the present study. Individuals were informed about the research and signed a consent form. This study complies with the Declaration of Helsinki and was approved by the Research Ethics Committee of the Institute of Health Sciences of the Federal University of Pará (CAAE 26765414.0.0000.0018 CEP-ICS/UFPA). All analyzed data were anonymized to protect the privacy of participants.

### 4.2. SNPs Selection

Twenty-five SNPs were selected based on association studies found in the PubMed database (www.ncbi.nlm.nih.gov/, accessed on 1 January 2020). These polymorphisms contribute to biological processes and immune responses in infectious diseases. Appendix A summarizes the main features related to the polymorphisms investigated in the present work.

### 4.3. DNA Extraction and Quantification

For each individual participating in the research, a blood sample was collected in 10mL tubes of the K2-EDTA type (Beckton Dickinson, Franklin Lakes, NJ, USA). The genetic material was extracted from the peripheral blood of the patients and the control group using the phenol-chloroform method, based on. DNA quantification was performed using NanoDrop 1000 Spectrophotometer equipment (NanoDrop Technologies, Wilmington, DE, USA).

### 4.4. Genotyping and Quality Control of Investigated SNPs

SNP Genotyping was performed by allelic discrimination using TaqMan technology. OpenArray Genotyping, with a panel of 32 custom assays on the QuantStudio ™ 12K Flex Real-Time PCR System (Thermo Fisher Scientific, Waltham, MA, USA) according to the manufacturer’s recommended protocol. The Taqman software Genotyper (Thermo Fisher Scientific, Waltham, MA, USA) was used to analyze plate data and genotype reading accuracy, in addition to genotyping quality control.

### 4.5. Ancestry Informative Markers

A panel of 61 ancestry informative markers (AIM) previously developed and expanded by our research group was used in this study, according to established protocols [80,81]. The AIMs panel is composed of specific ancestry markers capable of estimating individual and population genetic contributions to control possible ancestry influences.

### 4.6. Statistical Analysis

Genetic ancestry inference based on the AIM panel was performed in Structure software version 2.3.4 (Pritchard Lab, Stanford University, CA, USA) [82,83]. Differences in demographic and clinical characteristics such as age, sex, and ancestry analysis were compared using Student’s t-test, chi-squared and Mann–Whitney tests, respectively. Hardy–Weinberg equilibrium (HWE) and logistic regression analyses between leprosy genotypes and risk and its clinical forms PB and MB were performed by the SNPassoc version 2.0-11 package (Gonzáles et al., Barcelona, ESP) [84] with covariate adjustment. All statistical analyzes were performed using the statistical program R version 4.1.0 (Ross Ihaka & Robert Gentleman, Auckland, NZ) [85]. Values of *p* ≤ 0.05 were considered statistically significant.

## Figures and Tables

**Table 1 ijms-23-10628-t001:** Demographic and clinical characteristics of the sample of subjects.

Variables	Case (n = 114)	Control (n = 71)	*p*-Value
Age, years ^1^	39.62 ± 17	36.92 ± 18	0.324
Sex, % male/female ^2^	62 (54.4)/52 (45.6)	41 (57.7)/30 (42.3)	0.146
genetic ancestry ^3^			
European	0.640 ± 0.218	0.571 ± 0.251	**0.007**
Amerindian	0.199 ± 0.161	0.203 ± 0.166	0.882
African	0.161 ± 0.153	0.226 ± 0.203	**0.012**
**Variables**	**PB (n = 56)**	**MB (n = 58)**	
Age, years ^1^	36.34 ± 15	42.96 ± 19	**0.040**
Sex, % of male/female ^2^	19 (34.0)/37 (66.0)	43 (74.2)/15 (25.8)	**0.001**
genetic ancestry ^3^			
European	0.668 ± 0.190	0.652 ± 0.220	0.841
Amerindian	0.211 ± 0.170	0.181 ± 0.150	0.320
African	0.121 ± 0.130	0.167 ± 0.110	0.132

^1^ Values are expressed as mean ± SE (Standard Error of Mean), Student’s t-test; ^2^ Values are expressed as distribution percentages, chi-squared test; ^3^ Mann–Whitney test; The data are shown as mean ± standard deviation.

**Table 2 ijms-23-10628-t002:** Overview and comparison of the main results found according to the genotype associations between all the different groups studied. The values in bold show comparisons with a significant association.

Gene	Model	Case vs. Control	PB vs. Control	MB vs. Control	PB vs. MB
*pre-miR938* (rs2505901)	TT vs. TC + CC	**<0.01 ¹** **↓0.40 (0.20–0.81) ²**	**<0.01 ¹** **↓0.31 (0.12–0.75) ²**	0.07 ¹ 0.49 (0.22–1.08) ²	0.18 ¹ 1.90 (0.74–4.90) ²
	TT + CT vs. CC	**0.03 ¹** **↓0.41 (0.18–0.95) ²**	**0.01 ¹** **↓0.31 (0.12–0.84) ²**	0.21 ¹ 0.56 (0.22–1.42) ²	0.07 ¹ 2.56 (0.89–7.38) ²
*DROSHA* (rs639174)	CC vs. CT + TT	**0.02 ¹** **↓0.45 (0.23–0. 89) ²**	0.08 ¹ 0.50 (0.22–1.12) ²	**0.03 ¹** **↓0.43 (0.19–0.98) ²**	0.73 ¹ 0.85 (0.32–2.22) ²
*DROSHA* (rs10035440)	TT vs. CT + CC	**0.03 ¹** **↑2.04 (1.03–4.02) ²**	0.32 ¹ 1.50 (0.66–3.39) ²	**0.01 ¹** **↑2.88 (1.22–6.79) ²**	0.63 ¹ 0.80 (0.31–2.05) ²
*AGO1* (rs636832)	GG vs. GA + AA	**0.01 ¹** **↓0.45 (0.23–0.88) ²**	0.11 ¹ 0.53 (0.24–1.16) ²	**0.04 ¹** **↓0.45 (0.21–0.99) ²**	0.52 ¹ 1.33 (0.55–3.18) ²
*pri-let-7a1* (rs10739971)	GG vs. GA + AA	**0.02 ¹** **↑4.66 (1.17–18.6) ²**	**<0.01 ¹** **↑13.3 (1.82–90.0) ²**	0.23 ¹ 2.54 (0.53–12.3) ²	**<0.01 ¹** **↑5.21 (1.59–7.86) ²**
	GG + GA vs. AA	**0.04 ¹** **↑5.09 (0.95–7.45) ²**	0.09 ¹ 6.55 (0.53–8.90) ²	0.13 ¹ 3.95 (0.58–27.1) ²	0.55 ¹ 2.68 (0.11–6.46) ²
*miR146A* (rs2910164)	GG + GC vs. CC	0.66 ¹ 1.31 (0.22–2.59) ²	0.52 ¹ 0.65 (0.18–2.43) ²	0.66 ¹ 4.26 (0.47–38.5) ²	**0.04 ¹** **↓0.14 (0.01–1.36) ²**
*miR570* (rs4143815)	GG vs. GC + CC	0.73 ¹ 1.14 (0.73–2.73) ²	0.82 ¹ 1.10 (0.49–2.47) ²	**0.03 ¹** **↓0.45 (0.21–0.96) ²**	0.17 ¹ 1.79 (0.77–4.17) ²
*miR200C* (rs12904)	GG vs. GA + AA	**0.01 ¹** **↑2.77 (1.25–6.11) ²**	**<0.01 ¹** **↑3.42 (1.36–8.57) ²**	0.06 ¹ 2.28 (0.93–5.58) ²	0.26 ¹ 1.66 (0.68–4.06) ²
*miR4513* (rs2168518)	GG + GA vs. AA	0.38 ¹ 1.68 (0.53–5.34) ²	**0.03 ¹** **↑7.65 (0.83–70.5) ²**	0.87 ¹ 0.91 (0.27–3.03) ²	0.09 ¹ 5.37 (0.56–9.89) ²

¹ *p*-value; ² Odds Ratio and 95%CI; ↑ increased leprosy risk; ↓ decreased leprosy risk.

## Data Availability

Not applicable.

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
