# Peer review of "Association between SNPs in microRNAs and microRNAs-Machinery Genes with Susceptibility of Leprosy in the Amazon Population"

_ijms, 2022, doi:10.3390/ijms231810628_

Round 1
Reviewer 1 Report
Dear Authors,
I read your paper, being a clinician. My concern is: About 80% of humans will never develop leprosy. 20% are able to. Of this 20% the adaptive immunity protects, most about 90%, depending on genetics and history with previous antigenic contact and environmental circumstances.
So I wonder whether your control group is a real control group. How sure are you that they got an inoculum big enough and frequently enough to become infected and not becoming ill? Why not look at antiPGL1- positive not having the leprosy illness. Of them about 30% does get leprosy. Anti-PGL-1 Positivity as a Risk Marker for the Development of Leprosy among Contacts of Leprosy Cases: Systematic Review and Meta-analysis Penna et all. PLoS Negl Trop Dis. 2016 May; 10(5): e0004703.Published online 2016 May 18. doi: 10.1371/journal.pntd.0004703
When your contacts have no leprosy and are anti-PGL1 positive you at least know that they were infected.
I miss the socioeconomic status of patients and contacts and whether the contacts live in a high endemic area. Even in Acre, there are hotspots.
But the most problem I have with the sequence: Why not introduction, Material and methods then results and thereafter discussion and conclusions? You have Material and Methods after discussion.
Value of the results I can not judge because I do not know the consequences of an SNP in relation to infection. Not my field. It certainly does not look straightforward.
Some further remarks on the clinical part:
Line 40: it is caused by M. leprae and or M. lepromatosis.
Line 42: do not forget the sensory loss.
Line 46: The WHO recorded not detected.
Line 53: The broad spectrum is not the cause. But just leprosy has its classification based……
Line 154: do your SNP findings correlate with the findings about the ethnic groups and risk of leprosy?
General question: Tuberculoid patients may downgrade towards Lepromatous. Could that be caused by more SNP’s developing or being induced?
Reviewer 2 Report
In this article, the author tried to establish a link between SNPs in microRNAs and microRNAs-machinery genes with susceptibility to leprosy in the Amazon population.
Major comments:
- Clinical and demographic data collection reflects that men are more susceptible and have a higher frequency in the patient group than women. Is there any explanation or study reported for this statement? Is this observation made the first time in this study? If yes, is there any justification for this?
- The frequency of European ancestry in the patient group is higher compared to African ancestry. This observation complied with the research published by Pinto et al in 2015. In the present study, did the authors try any of these genes as a control to confirm the outcomes?
- The frequency of African ancestry was higher in healthy individuals and these control cases were in close relation to the patient groups. How are these samples related to each other, are they from the same lineage, if yes, then how come they are less susceptible to leprosy?
- Are there any unrelated healthy individual cases used in this study as a negative control?
- In table 1 American Indian case have a high standard error. Is there any specific reason for this observation?
- There are some calculation errors in table 1 need to be corrected.
Minor comments:
- In line 52, italicized the scientific name.
- In line 72, provide some examples of biological processes to complete this sentence.
- Typing error in line 121.
- The BP form used in lines 203 and 242 is not defined in the manuscript.
- In table 2, please provide a simpler way to express the association with increased or decreased risk for leprosy (like arrows in an upward or downward direction). Currently, all values are in bold, and without reading the corresponding text it's difficult to understand the table. This could be done with some additional text in the table's legend.
- Please see attached file for more details.

Round 2
Reviewer 1 Report
Dear Authors,
I thank you for answering my questions. My excuses concerning the sequence, I should have been aware.
Some additional comments : Both anti-PGL-1 and qPCR may miss paucibacillary patients as you showed in your own cited paper. But you have decreased clinically the number that may have been missed among the control group. I can certainly agree with it.
Some mall remarks:
In table 1 the age of MB patients is higher than among the PB is because the development of MB seems to cost more time. The downgrading and symptoms may develop later.
Line 282: add the answer to the other reviewer: the control group are not blood relatives of the patients in the patient group.
Author Response
Dear reviewer 1,
we would like to thank you again for your time committed to improving our manuscript. Below are the answers to your questions.
Some mall remarks:
Point 1: In table 1 the age of MB patients is higher than among the PB is because the development of MB seems to cost more time. The downgrading and symptoms may develop later.
Response 1: Dear reviewer, thank you for the information. We have chosen to insert a short sentence about this clinical aspect of the disease for a better understanding of the readers (line 97)
Point 2: Line 282: add the answer to the other reviewer: the control group are not blood relatives of the patients in the patient group.
Response 2: Thanks for your observation. We have added your suggestion to the manuscript (line 283).
Reviewer 2 Report
The author provided sufficient details and explanation for my comments.
Author Response
Dear Reviewer 2,
We would like to thank you for taking the time to improve our manuscript.